# The Equine Faecal Microbiota of Healthy Horses and Ponies in The Netherlands: Impact of Host and Environmental Factors

**DOI:** 10.3390/ani11061762

**Published:** 2021-06-12

**Authors:** Mathijs J. P. Theelen, Roosmarijn E. C. Luiken, Jaap A. Wagenaar, Marianne M. Sloet van Oldruitenborgh-Oosterbaan, John W. A. Rossen, Aldert L. Zomer

**Affiliations:** 1Department of Clinical Sciences, Faculty of Veterinary Medicine, Utrecht University, Yalelaan 112, 3584 CM Utrecht, The Netherlands; m.sloet@uu.nl; 2Department of Infectious Diseases and Immunology, Faculty of Veterinary Medicine, Utrecht University, Yalelaan 1, 3584 CL Utrecht, The Netherlands; r.e.c.luiken@uu.nl (R.E.C.L.); j.wagenaar@uu.nl (J.A.W.); a.l.zomer@uu.nl (A.L.Z.); 3Department of Medical Microbiology and Infection Prevention, University Medical Center Groningen, University of Groningen, Antonius Deusinglaan 1, 9713 AV Groningen, The Netherlands; j.w.a.rossen@rug.nl; 4Department of Pathology, University of Utah School of Medicine, Salt Lake City, UT 84132, USA

**Keywords:** equine, faecal, microbiota, age, gender, pony, diet, pasture, season, location

## Abstract

**Simple Summary:**

Several studies have described the bacterial composition in the intestines of horses, and several factors of influence have been detected. Variation in the results between studies, however, is substantial. Therefore, the current study aimed to study the bacterial composition in the faeces of healthy horses and ponies kept under standard housing and management condition in The Netherlands. Seventy-nine horses and ponies originating from two farms were included. Several factors, such as location, age, the season of sampling, horse type (horses vs. ponies) and pasture access significantly affected the bacterial composition. The current study provides important baseline information on variation in the bacterial composition in healthy horses and ponies under standard housing and management conditions. The aforementioned factors identified in this study to affect the bacterial population of the gut should be considered in future studies regarding the bacterial population of the equine gut.

**Abstract:**

Several studies have described the faecal microbiota of horses and the factors that influence its composition, but the variation in results is substantial. This study aimed to investigate the microbiota composition in healthy equids in The Netherlands under standard housing and management conditions and to evaluate the effect of age, gender, horse type, diet, pasture access, the season of sampling and location on it. Spontaneously produced faecal samples were collected from the stall floor of 79 healthy horses and ponies at two farms. The validity of this sampling technique was evaluated in a small pilot study including five ponies showing that the microbiota composition of faecal samples collected up to 6 h after spontaneous defaecation was similar to that of the samples collected rectally. After DNA extraction, Illumina Miseq 16S rRNA sequencing was performed to determine microbiota composition. The effect of host and environmental factors on microbiota composition were determined using several techniques (NMDS, PERMANOVA, DESeq2). Bacteroidetes was the largest phylum found in the faecal microbiota (50.1%), followed by Firmicutes (28.4%). Alpha-diversity and richness decreased significantly with increasing age. Location, age, season, horse type and pasture access had a significant effect on beta-diversity. The current study provides important baseline information on variation in faecal microbiota in healthy horses and ponies under standard housing and management conditions. These results indicate that faecal microbiota composition is affected by several horse-related and environment-related factors, and these factors should be considered in future studies of the equine faecal microbiota.

## 1. Introduction

A well-functioning intestinal tract and intestinal microbiota are considered essential for maintaining health in horses [1]. Disturbances of the microbiota are associated with diseases in horses, such as colitis, equine metabolic syndrome and colic [2,3,4], although it is difficult to assess what comes first, disease or altered microbiota. Several studies have described the faecal microbiota of healthy horses, but the variation in results is substantial [5,6,7,8,9,10,11,12,13,14,15,16,17]. Therefore, our understanding of what can be considered normal variation and truly abnormal is currently limited. Inter-individual variation has been demonstrated to be larger than intra-individual variation [7,18]. The single main predictor of microbiota composition is individual identity, and it was suggested that this explains about 50% of the variation [19], meaning that other factors also affect microbiota composition. Several research groups studied the effects of different factors such as age, breed, band, maternal relationship, social behaviour, environmental conditions (such as diet, pasture access, fasting, transportation, exercise and season) and the use of pre- and probiotics and antimicrobials on the equine gut microbiota [6,8,9,10,13,14,15,19,20,21]. Geographic variation in microbiota composition has been demonstrated in humans [22], and the same might be true for horses, highlighting the need for studies from different geographic regions assessing faecal microbiota composition in horses. So far, most studies have compared relatively homogenous groups of horses exposed to one changing variable. This has the advantage of detecting subtle differences attributable to the tested variable, but at the same time, limits the extent to which results can be extrapolated to other populations of horses managed differently. Therefore, more knowledge about the normal faecal microbiota of horses and ponies kept under standard housing and management conditions is needed, for example, to study potential associations between microbiota composition and disease status. This study aimed to describe the microbiota composition in healthy horses and ponies in The Netherlands kept under standard housing and management conditions and assess which factors influence faecal microbiota composition.

## 2. Materials and Methods

### 2.1. Study Design, Population and Metadata Collection

The horses and ponies (*n* = 79) included in the study were client-owned animals housed at two locations in The Netherlands and sampled once between April 2015 and February 2016. At the main location, farm I, 61 animals (aged 5–31 years) were included. All animals were kept in individual stables with straw bedding, and some had pasture access. An additional 18 Warmblood horses (age 4–16 years) were sampled at a second farm (II). These horses were also kept in individual stables on straw or sawdust and some of the horses had pasture access. For each animal, information regarding age (in years), gender (male/female), horse type (horse/pony), diet (type of roughage and type and amount of concentrates), pasture access (yes/no) and season of sampling (summer/winter) was recorded. The minimum and maximum environmental temperatures at the moment of sample collection in the summer were 8 to 18 °C and 3 to 10 °C in the winter. None of the animals included in the study were treated with antimicrobials within the last six months, and none of the horses had any health problems in the past six months, according to the owner.

### 2.2. Ethical Considerations

No procedures had to be performed on the animals included in this study. Therefore, ethical approval was not required for this study. Informed consent was obtained from all the owners.

### 2.3. Faecal Sampling

Faecal samples were collected from the stall floor of individually housed horses and ponies. To ensure fresh samples were collected, sampling occurred within 6 h after the stables had been cleaned. Faeces were collected from the centre of a faecal ball. The samples were stored at −80 °C within 2 h of collection. To evaluate the validity of the sampling technique described above, a pilot study was conducted to investigate the effect of air exposure on the equine faecal microbiota when using this convenience sampling technique. For this purpose, faecal samples were collected rectally (*t* = 0) from five Shetland ponies (that underwent a rectal exam for reproductive purposes), after which these samples were exposed to room air (temperature: 18.2–23.9 °C; humidity 55.0–68.0%) and aliquots were sampled at 1 h, 3 h, 6 h, 12 h and 24 h.

### 2.4. DNA Extraction & 16S rRNA Amplicon Sequencing

DNA extraction was performed using the QIAamp Fast DNA Stool Mini Kit (Qiagen, Hilden, Germany) following a previously published protocol [23]. However, the samples were not treated in the TissueLyser at 30 Hz for 3 × 30 s with cooling on ice in between treatments but were bead-beaten for 5 min on a Vortex-Genie 2 (Scientific Industries, Bohemia, NY, USA). The variable V3 and V4 regions of the 16S rRNA amplicon were amplified, and libraries were prepared following the 16S Metagenomic Sequencing Library Preparation protocol (Illumina, San Diego, CA, USA). Next, each library was normalised, pooled and loaded onto the Illumina MiSeq platform for paired-end sequencing using the 600 cycles MiSeq Reagent Kit V3 (Illumina, San Diego, CA, USA), generating 2 × 300 base pair paired-end reads.

### 2.5. Bioinformatics Processing

Data preparation was performed using Jupyter notebook version 5.7.8, running on Python 3.7.3. utilising R version 3.4.4. Raw reads (250 bp) obtained from Illumina 16S rRNA amplicon sequencing provided input for the denoising pipeline DADA2. DADA2 models and corrects Illumina-sequenced amplicon errors with high precision [24]. First, the forward and reverse reads were sorted, and the quality profile was plotted. Trimming parameters were derived from the quality plots, maintaining a minimum quality score of 20. Forward-reads contained higher quality compared to reverse reads, as is common among Illumina data. Truncations were set at 15–290 for forward, 15–210 for reverse. Post filter and trimming the reads were merged and the merged data was used to create a sequence table. After the removal of chimeras, taxonomy was assigned using v. 132 of the Silva database [25].

### 2.6. Data Analysis

The DADA2 object was imported into the Phyloseq package [26]. All samples with less than 5000 reads were excluded. All data analyses and visualisations were performed with R version 4.0.2 [27] using vegan [28] and ggplot [29] packages. For alpha-diversity, data was rarefied to the sample with the lowest read counts (14,665 reads). All analyses were performed on the data of 61 horses and ponies from farm I unless mentioned otherwise.

#### 2.6.1. Relative Abundance and Alpha-Diversity

Relative abundances at the phylum level were assessed for each sample, and phylum and class level bar plots were produced. Alpha-diversity (observed richness and Shannon diversity) was calculated from rarefied data. The effect of potential determinants (age, gender, horse type, roughage, concentrates, pasture access, season and location) on sample alpha-diversity was univariably tested with Wilcoxon rank-sum test (two groups), Kruskal–Wallis test (multiple groups) or linear regression (continuous variable). To evaluate the effect of location (farm) on the faecal microbiota, horses of farm II were compared to a comparable group of 17 Warmblood horses in the same age group from farm I.

#### 2.6.2. Microbiota Composition (Beta-Diversity)

Between sample Bray–Curtis dissimilarity was computed on relative abundance data and used for Non-Metric Dimensional Scaling (NMDS). To determine if significant differences in microbiota composition were present between groups based on the previously mentioned factors, permutational multivariate analysis of variance (PERMANOVA), including beta-dispersion analysis, was performed (vegan package function Adonis2 and betadisper).

#### 2.6.3. Differential Abundance Analysis

To determine taxa that differ in abundance for the tested factors, the DESeq2 package [30] was used. With a Wald test, the DESeq2 package determines if a significant fold change is present. The *p*-values were adjusted for the FDR (Benjamini–Hochberg approach [31]) with alpha set at 0.05. For this analysis, only factors associated with a significant difference in beta-diversity were considered. Raw count data was used as input, taxa not seen more than three times in at least 20% of the samples were filtered beforehand. Percentages of interesting taxa were calculated to reveal abundances in the entire microbiome.

## 3. Results

Faecal samples from 79 equids were included in the study. Sixty-one animals located at farm I were included in the analysis to study the effects of age, gender, horse type, roughage, concentrates, pasture access and season on faecal microbiota composition. See Table 1 for descriptive characteristics of the study population. Eighteen horses originating from farm II were compared to 17 horses of the same breed and age from farm I to study the effect of location/farm on faecal microbiota composition. A total of 5,015,145 high quality non-chimeric bacterial 16S reads were generated and annotated to 25,011 OTUs. The distribution of the number of reads across samples was as follows: a minimum of 14,665 reads, a maximum of 87,169 reads and a median of 32,949 reads.

### 3.1. Effect of Air Exposure on the Equine Faecal Microbiota

To evaluate the validity of the used sampling technique of collecting fresh faecal samples from the stall floor, a pilot study was conducted to investigate the effect of air exposure on the equine faecal microbiota. Faecal balls were collected rectally from 5 ponies and exposed to room air for different times. Richness and alpha-diversity were stable up to 12 h, but decreased between 12 and 24 h of air exposure, see Figure 1. Significant shifts in relative abundance for different phyla were observed at *t* = 12 h and *t* = 24 h compared to *t* = 0. Bacteroidetes decreased in relative abundance, while Firmicutes increased. At the class level, Bacilli increased after 12 h of air exposure (Appendix A).

### 3.2. Faecal Microbiota Composition

A relative abundance of taxa in the faecal microbiota of 79 horses and ponies at class level and phylum level is presented in Appendix A. Phyla, classes and families with a relative abundance of >1% are presented in Table 2. Bacteroidetes was the phylum with the largest mean relative abundance in our study (50.1%), followed by Firmicutes (28.4%), Spirochaetes (7.1%), Verrucomicrobia (6.5%), Fibrobacteres (5.0%) and Cyanobacteria (1.0%).

### 3.3. Factors Affecting Faecal Microbiota Composition

We evaluated the effects of age, gender, horse type, diet (roughage type, concentrates), pasture access, the season of sampling and location on microbiota composition by comparing the alpha-diversity, beta-diversity and differential abundance of taxa between samples.

#### 3.3.1. Age

Increasing age led to decreased observed richness (*p* < 0.001) and alpha-diversity (*p* = 0.015), see Figure 2. Age also significantly affected beta-diversity, suggesting that the microbiota composition changes as animals age (R^2^ = 0.031, *p* = 0.002). Several amplicon sequence variants (ASVs) were significantly less abundant in older horses, including ASVs assigned to the bacterial families Acidaminococcaceae, Ruminococcaceae, p-251-o5 and an unidentified taxon belonging to the phylum of Bacteroidetes, see Figure 3. Other ASVs were more abundant in older horses. These were assigned to the families Eggerthellaceae, Lactobacillaceae, Selenomonadaceae, Oscillospiraceae, Erysipelatoclostridiaceae, Ruminococcaceae, Anaerovoracaceae, Rikenellaceae, Lachnospiraceae, Prevotellaceae, Spirochaetaceae, Bacteroidales_UCG-001, F082 and UCG-010.

#### 3.3.2. Gender

No significant differences in observed richness, alpha- or beta-diversity were observed for gender in this study.

#### 3.3.3. Horse Type

Thirty-two horses (Dutch Warmblood, Rheinlander, Oldenburger and Standardbred horses) and 29 ponies (Haflinger, Tinker, Irish Cobs, Welsh, Appaloosa, New Forest, Fjord, Icelandic, Shetland and mini-Shetland ponies) were included to evaluate for differences in faecal microbiota between horses and ponies. No significant differences in observed richness and alpha-diversity were observed for horses compared to ponies. Beta-diversity was significantly different when horses and ponies were compared. Horse type determined 2.4% of the variation (PERMANOVA *p* = 0.015 betadisper *p* = 0.530). ASVs assigned to the following families of bacteria were significantly less abundant in ponies compared to horses: Ruminicoccaceae, Lactobacillaceae, Prevotellaceae, Acidaminococcaceae, Rikenellaceae, p-251-o5 and some unidentified taxa within the Firmicutes phylum. Other ASVs were more abundant in ponies compared to horses. These belonged to the bacterial families of Lachnospiraceae, Acidaminococcaceae, Saccharimonadaceae, Rikenellaceae, Spirochaetaceae and p-251-o5, see Figure 4.

#### 3.3.4. Diet

No significant differences in observed richness, alpha- or beta-diversity were observed for the type of roughage fed. Also, no significant differences in richness, alpha- or beta-diversity were observed for animals fed more than 2 kg of concentrates compared to animals fed less than 2 kg of concentrates.

#### 3.3.5. Pasture Access

No significant differences in observed richness and alpha-diversity were observed for pasture access. However, a significant difference in beta-diversity was observed when horses with pasture access were compared to horses that did not have pasture access. Pasture access explained 2.3% of the variation in microbiota composition (PERMANOVA *p* = 0.035 betadisper *p* = 0.034). Only a few ASVs were significantly less abundant in animals with pasture access. These belonged to the family of Lachnospiraceae and an unidentified taxon within the phylum of Bacteroidetes. Other ASVs were more abundant in animals with pasture access, belonging to the bacterial families of Prevotellaceae, Rikenellaceae, UCG-010 and two unidentified taxa within the phylum of Bacteriodetes.

#### 3.3.6. Season

No significant differences in observed richness and alpha-diversity were identified for the season of sampling. However, the microbiota composition of samples collected in summer differed from that of samples collected in winter, evidenced by a significant difference in beta-diversity. The season of sampling explained 2.8% of the variation (PERMANOVA *p* = 0.002, betadisper *p* = 0.589). Firmicutes were significantly less abundant in samples collected in summer, Bacteroidetes were significantly less abundant in samples collected in winter (Figure 5). On a more detailed level, several ASVs were significantly less abundant in samples collected in the summer than samples collected in the winter. These were assigned to the bacterial families Planococcaceae, Anaerovoraceae, Oscillospiraceae, Rikenellaceae, Lachnospiraceae, Erysipelatoclostridiaceae, F082, Bacteroidales_UCG-001, p-251-o5 and several unidentified taxa within the phyla of Bacteroidetes, Firmicutes and Cyanobacteria. Other ASVs were more abundant in samples collected in the summer than samples collected in the winter, belonging to the bacterial families Ruminococcaceae, Rikenellaceae, Desulfovibrionaceae, Synergistaceae, Lachnospiraceae, Erysipelotrichaceae, Erysipelatoclostridiaceae, Prevotellaceae, Spirochaetaceae and unidentified taxa within the Firmicutes phylum; see Figure 6.

#### 3.3.7. Location

To evaluate the effect of location (farm) on microbiota composition, data from 17 Warmblood horses aged 4 to 16 years from farm I were compared to data from 18 Warmblood horses of the same age from farm II. No significant differences in observed richness and alpha-diversity were seen for horses from farm I compared to horses from farm II. However, apparent clustering of samples from horses according to farm/location can be observed in the NMDS plot, see Figure 7, explaining 6.4% of the variation (PERMANOVA *p* = 0.001, betadisper *p* = 0.779).

## 4. Discussion

The equine faecal microbiota, as well as factors that shape it, have been studied and described in several publications in recent years but the reported results vary significantly [5,6,7,8,9,10,11,12,13,14,15,16,17]. These differences might, in part, be a result of methodological differences such as DNA extraction and sequencing techniques used. The large differences in reported results regarding the composition of the intestinal microbiota of healthy horses reduces our ability to draw conclusions from studies evaluating associations between specific equine diseases or physiological factors and the intestinal microbiota. The many uncertainties regarding the intestinal microbiota of healthy horses that still exist today support the publication of new studies evaluating the faecal microbiota of healthy subjects and factors that influence its composition.

### 4.1. Effect of Air Exposure on the Equine Faecal Microbiota

To validate the use of collecting spontaneously produced faecal samples as a form of convenience sampling in the current study, we studied the effect of air exposure on microbiota composition. Our results show that the observed richness and the α-diversity of the microbiota in faecal samples remain stable for up to 6 h, after which they decrease at 12 h of air exposure. The microbiota composition of faecal samples collected up to 6 h after spontaneous defaecation is the same as that of samples collected rectally. Therefore, these samples can be used for faecal microbiota analysis, which validates the faecal sample collection technique applied in the current study. This corresponds to previous reports in which parameters were also reported to be stable up to 6 h after defecation [17,32]. In our study, we found the relative abundance of Bacteroidetes to decrease and Firmicutes to increase over time. At the class level, Bacilli increased significantly in our study, which is in agreement with another study that identified Bacillaceae, Planococcaeae and Enterococcaceae, all families within the class of Bacilli, as bloom taxa [32].

### 4.2. Faecal Microbiota Composition

In the current study, Bacteroidetes was the most abundant phylum in the faecal microbiota, followed by Firmicutes, Spirochaetes and Fibrobacteres. Our results closely resemble those of some other studies in horses [7,14,17,19]. However, Firmicutes is the main bacterial phylum found in most studies of the faecal microbiota in healthy horses to date [2,3,5,6,8,9,10,12,13,15,16,21,33,34,35,36]. The relative abundance of Bacteroidetes in those studies varies significantly, ranging from 0% to 42% (mean 19%) [3,13]. Of the studies reporting high levels of Bacteroidetes (range 33–52%; mean 43%), similar to our results, almost all used the same DNA extraction kit (QiaAmp Fast DNA Stool Mini Kit) and protocol [7,8,13,14,19,35]. Studies using another commonly used DNA extraction kit and protocol (E.N.Z.A. Stool DNA Kit, Omega Bio-tek, Norcross, GA, USA) consistently report a lower relative abundance of Bacteroidetes ranging from 0% to 25% (mean 7%) [2,3,5,10,15,21]. This indicates that DNA extraction kit and protocol might be a major factor of influence on the results in equine faecal microbiota studies. Several studies have shown significant effects of DNA extraction methods (and sample type) on the results of microbiota analysis [23,37]. However, a study comparing the two DNA extraction kits most commonly used in equine microbiota studies, as mentioned above, is currently lacking. The use of different 16S primers for sequencing might also affect results [38]. Studies evaluating the effect of 16S primer selection on results in equine studies are also currently lacking. The large effect of several critical steps in obtaining microbiota data, as mentioned above, complicates the comparison of taxa abundance and microbiota composition between studies when different protocols are used.

### 4.3. The Effect of Different Factors on Faecal Microbiota Composition

#### 4.3.1. Age

Our results show a reduction in the number of species (observed richness) and alpha-diversity with the increasing age of horses. The current reports in the literature on horses are ambiguous on this topic. Similar results were obtained in a study comparing the faecal microbiota of mature horses (5–12 years) to that of elderly horses (19–28 years) [8]. In other studies, no effect of age on alpha-diversity was observed [11,39] and in specific breeds (Anglo-Arabian horses and wild Prezwalski horses), the opposite trend was observed where alpha-diversity increased with age [10,40]. In adult humans, alpha-diversity decreases with age, similar to the trend observed in our study in horses [41]. In our study, we identified several ASVs whose relative abundance changed with increasing age. Little is known about the role these taxa play in the gut and how this might affect host health. Of interest is the decrease of two taxa within the family of Ruminococcaceae with increasing age in our study, as Ruminococcaceae have been associated with gut health in mammals and lower relative abundance has been associated with colitis, diarrhoea, colic and equine metabolic syndrome in equids [2,4,36,42]. Older horses, therefore, might be more prone to the development of specific diseases. In humans, increasing age, especially in the group of centenarians, leads to a compromised microbiota with increased levels of pathobionts and a heightened inflammatory state, known as inflammageing, and an increase in peripheral blood inflammatory markers can be observed. [41] If the same process occurs in horses is currently unknown. A marked decrease in *Faecalibacterium prauznitzii* (a member of the Ruminococcaceae family) was observed in humans. Our study also observed a decrease of specific ASVs assigned to two families of Ruminococcaceae in horses with increasing age, but we were unable to further specify genus/species.

#### 4.3.2. Gender

In the current study, no effect of gender was found on the faecal microbiota. This is in line with reports from other studies in horses and ponies [5,19]. However, one study did report gender differences in Przewalski horses kept under similar circumstances [43]. Gender differences in faecal microbiota are also observed in humans [44]. Gender differences in microbiota composition in horses might be subtle.

#### 4.3.3. Horse Type

Horses and ponies differ in metabolism, and as a result, ponies are more prone to obesity and related disorders such as laminitis and equine metabolic syndrome [45,46,47]. Differences in the prevalence of several gastrointestinal disorders have also been described for horses and ponies [48]. Therefore, we evaluated differences in microbiota composition between horses and ponies in our study. We found no differences in alpha-diversity between horses and ponies, which is in line with a recently published study [49]. However, a significant difference in beta-diversity was observed when horses and ponies were compared, demonstrating significant differences in microbiota composition. Ruminococcaceae (fibrolytic bacteria), Lactobacillaceae (carbohydrate fermentation, production of lactic acid) and Prevotellaceae (breakdown of carbohydrates and protein) were less abundant in ponies compared to horses. Lachnospiraceae (plant polysaccharide fermentation and short-chain fatty acid production), Saccharimonadaceae and Spirochaetaceae were more abundant in ponies. For Acidaminococcaceae, Rikenellaceae and P-251-o5, we observed that specific genera/species within these families increase or decrease according to horse type, suggesting that bacterial shifts occur within these families. In the study by Langner et al. (2020), Ruminococcaceae, Rikenellaceae, F082 and Bacteroidales UCG-001 were found to be more abundant in ponies compared to horses, but only after all subjects included in the study were kept on a 200% net energy requirement diet for two years [49]. The differences, therefore, not only reflect differences between ponies and horses, as they were not present at the start of the study but are heavily influenced by the high energy diet. Breed differences are also described in other studies comparing the microbiota of several European sport horse breeds and Thoroughbred horses and Mongolian horses [10,35]. Most of the identified differences were subtle or were seen only in low abundant phyla and genera. A study comparing the microbiota composition of Quarter Horses, Morgan, Paint and Tennessee Walking found no differences [50]. The differences in microbiota composition between horses and ponies observed in the current study indicate that horse type should be considered in experimental studies of the faecal microbiota.

#### 4.3.4. Diet

Our study did not observe a significant effect of diet (roughage and/or concentrates) on microbiota composition. This contrasts with previous studies in which dietary effects on the intestinal microbiota have been demonstrated [51,52,53]. However, in these studies, larger contrasts in diet between studied groups were tested (e.g., no concentrate feeding vs. high level of concentrate feeding). In our study, horses and ponies living under normal circumstances at a farm were sampled and, therefore, for example, received varying amounts and varying types of concentrates (with varying levels of nutrients) leading to less contrast when groups were compared and limiting our ability to detect differences. Also, the studies mentioned above evaluated the effect of a sudden change in diet instead of assessing the microbiota from equids on a long-term stable diet. Sudden changes in diet might disrupt the microbiota composition and therefore lead to more pronounced differences that are not present while on a steady diet, as was the case for the horses and ponies included in our study.

#### 4.3.5. Pasture Access

In contrast to roughage and concentrates, pasture access did significantly affect microbiota composition in the current study. Lachnospiraceae were less abundant in horses with pasture access in our study, whereas Prevotellaceae, Rikenellaceae and UCG-010 were more abundant. In a study performed in New Zealand, specific taxa were identified to be more abundant in horses abruptly transitioned to pasture: an unclassified genus within the order of Clostridiales, an unclassified genus within the order of Lachnospiraceae, CF231 and BF311 [9]. Another recent study reported an increase in Lactobacillaceae in horses that were abruptly changed from a hay-only diet to a grass diet [54]. The horses included in our study did not undergo abrupt dietary changes and had the same level of pasture access over a prolonged period prior to sampling, which might explain the differences observed when the results of our study are compared to that of previous reports. A recent study from Switzerland also detected significant differences in faecal microbiota composition depending on the number of hours horses had pasture access [55]. Based on the current study and the previously published data, pasture access seems to affect microbiota composition in equids.

#### 4.3.6. Season

We observed a seasonal effect on the faecal microbiota in our study, as evidenced by differences in beta-diversity. Genera/species within the same families increased or decreased with the season of sampling, suggesting that shifts occur within bacterial families. We observed this trend for Rikenellaceae, Lachnospiraceae and members of the family of Erysipelatoclostridiaceae. In a longitudinal study by Salem et al., a biphasic change in microbiota composition was observed over a 12-month period, and alpha-diversity showed a significant non-linear trend [14]. In our study, we did not observe any differences in alpha-diversity. In our study and the study by Salem et al., pasture access varied according to season and was higher in the summer months. Therefore, it is difficult to determine which of the observed differences resulted from dietary changes because of pasture access and which of the differences resulted from the change in season. Nevertheless, our results suggest that season should be included as a co-variant in future studies.

#### 4.3.7. Location

We demonstrated an effect of farm/location on microbiota composition in our study. A previously published study showed relatively low variation in microbiota composition between horses housed in the same environment, indicating interaction patterns across horses housed in the same environment [51]. The effects of location on microbiota composition have been reported previously in domesticated and (semi-) feral horses [19,34,39,56,57]. However, these differences might also in part be explained by dietary or management differences. The location effect should be included in future studies concerning the equine faecal microbiota.

### 4.4. Strengths and Limitations

As demonstrated in the current study and previous studies, many factors contribute to shaping the gut microbiota in individual animals. However, explained variation by individual factors is generally low, in our study up to 6.4%, suggesting that other (unidentified) factors also play a role. This complexity of involved (and potentially interrelated) determinants is a challenge for suitable study designs. The current broad assessment of the faecal microbiota in horses and ponies living under standard housing conditions at equine farms has the advantage that the results represent the actual situation in the field. The difficulty with a non-homogenous study population is that differences between groups of horses might be less pronounced than in experimental studies in a more homogenous study population with only one variable being studied. By including a more significant number of equids than most equine studies, we have tried to overcome this limitation. In the current study, horses from only two farms were included, limiting the extent to which results can be extrapolated to other equine populations. However, given the relatively large number of equids included in the study and the variation in horse characteristics and management conditions it does provide important baseline information on variation in faecal microbiota in healthy horses and ponies under field conditions. Larger, and preferably international, population-based studies are needed to further investigate and understand the complexity of the equine faecal microbiota and the factors that shape it.

## 5. Conclusions

The current study provides essential baseline information on variation in faecal microbiota in a group of horses and ponies managed under normal management circumstances and the factors that influence its composition. Bacteroidetes is the largest phylum found in the faecal microbiota of the horses and ponies included in this study, followed by Firmicutes. Alpha-diversity and richness decreased significantly with increasing age. Location, age, season, horse type and pasture access had a significant effect on the composition of the microbiome, explaining 2.3% to 6.4% of the observed variation. These results indicate that faecal microbiota composition is affected by several horse-related and environment-related factors. Studies investigating potential relationships between environmental factors or disease status and faecal microbiota should take these factors into account when interpreting observed differences to avoid the risk of overinterpretation.

## Figures and Tables

**Figure 1 animals-11-01762-f001:**
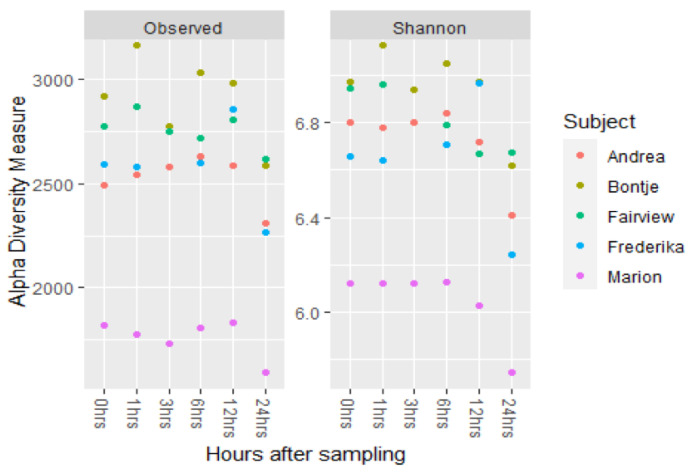
Observed richness and alpha-diversity (Shannon diversity index) of the faecal microbiota over time of faecal samples collected from five ponies after exposure to room air. A decrease in observed richness alpha-diversity is visible after 12 h of air exposure (*p* = 0.062).

**Figure 2 animals-11-01762-f002:**
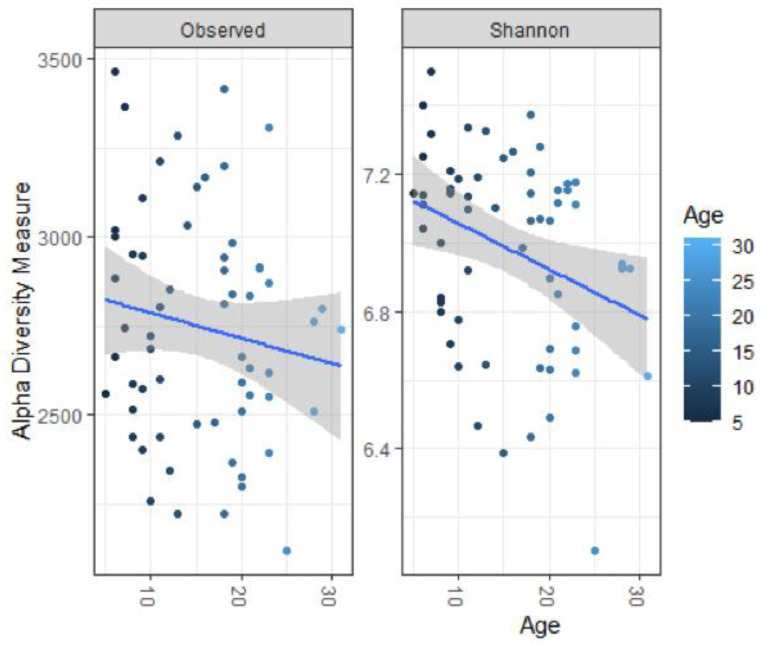
The effect of age on the observed richness and alpha-diversity (Shannon diversity index) on the faecal microbiota of 61 horses and ponies. A significant decrease, determined with linear modelling, in observed richness and alpha-diversity is visible in older horses.

**Figure 3 animals-11-01762-f003:**
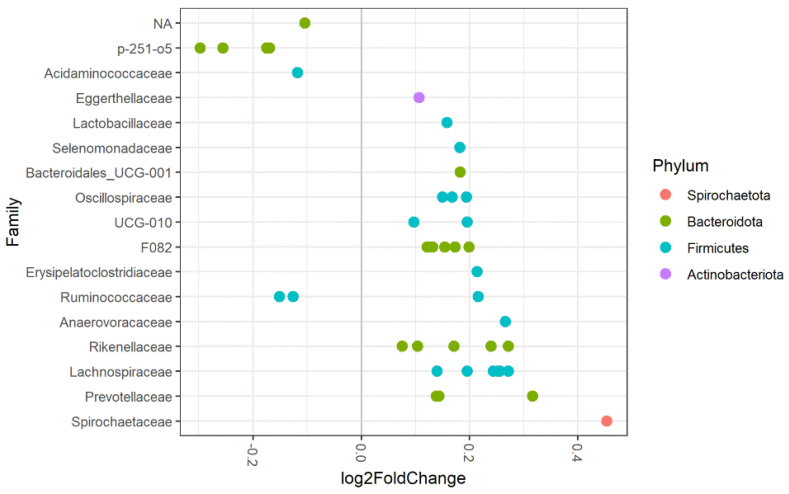
Differentially abundant ASVs grouped by family for increasing age (by year) of the faecal microbiota in 61 horses and ponies. The log2 fold change (per year) in ASV abundance is shown on the *x*-axis. ASVs assigned to bacterial families on the left side of the plot are less abundant in horses with increasing age. ASVs assigned to families depicted on the right side of the plot are more abundant in horses with increasing age. NA = ASV belonging to an unknown family (colours indicate the phylum).

**Figure 4 animals-11-01762-f004:**
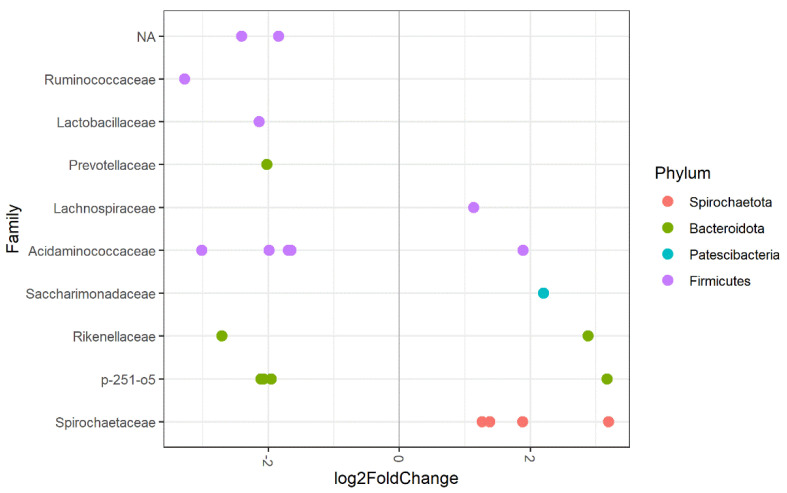
Differentially abundant ASVs grouped by family for horse type (ponies vs. horses) of the faecal microbiota in 61 horses and ponies. The log2 fold change in species abundance is shown on the *x*-axis. ASVs assigned to bacterial families on the left side of the plot are less abundant in ponies than horses, ASVs assigned to families depicted on the right side of the plot are more abundant in ponies than horses. NA = ASV belonging to an unknown family (colours indicate the phylum).

**Figure 5 animals-11-01762-f005:**
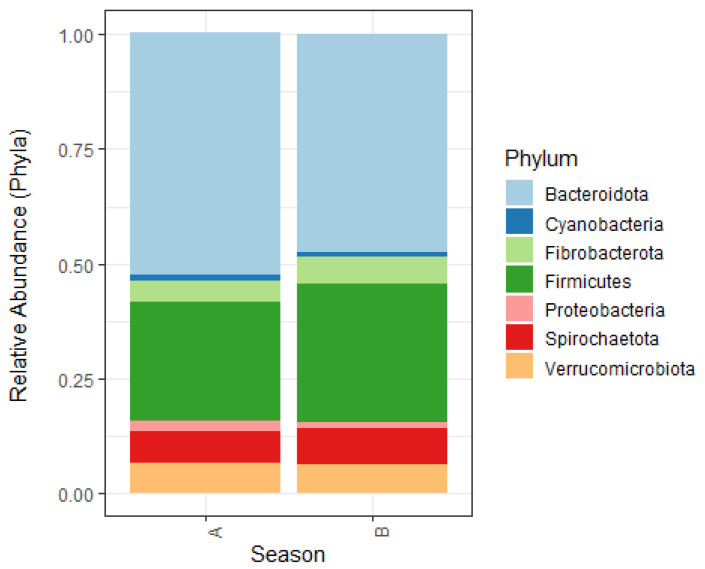
Relative abundance of phyla in the faecal microbiota of 61 horses and ponies in The Netherlands. (**A**) Summer (**B**) Winter.

**Figure 6 animals-11-01762-f006:**
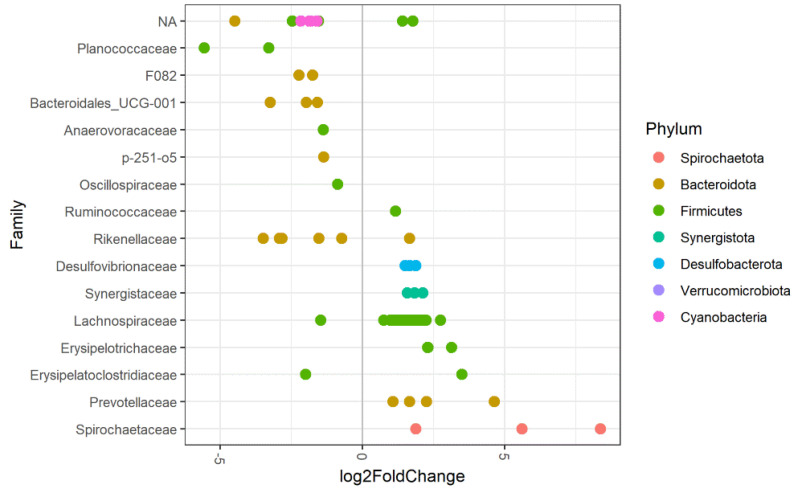
Differentially abundant ASVs grouped by family for season of sampling (winter vs. summer) of the faecal microbiota in 61 horses and ponies. The log2 fold change in species abundance is shown on the *x*-axis. ASVs assigned to bacterial families on the left side of the plot are less abundant in samples collected in winter compared to samples collected in summer, ASVs assigned to families depicted on the right side of the plot are more abundant samples collected in the winter compared to sampled collected in the summer. NA = ASV belonging to an unknown family (colours indicate the phylum).

**Figure 7 animals-11-01762-f007:**
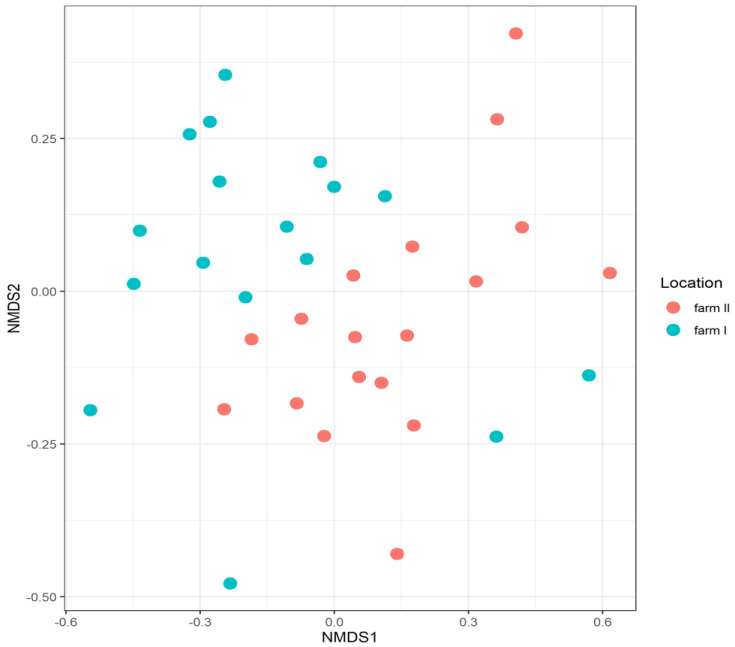
NMDS plot of beta-diversity showing the effect of location (farm) on the faecal microbiota composition of 35 horses. Red = Farm II, Blue = Farm I. NMDS stress level = 0.17.

**Table 1 animals-11-01762-t001:** Descriptive data of the study population.

Age	*n*	Mean (Years)	SD (Years)	Range (Years)
	61	15.6	6.7	5–31
Gender				
Male	37			
Female	24			
Horse type				
Pony	29			
Horse	32			
Roughage				
Hay	10			
Haylage	33			
Mixed	18			
Concentrates				
<2 kg	25			
≧2 kg	36			
Pasture access				
None	20			
Daily	41			
Season of sampling				
Summer	30			
Winter	31			
Location ^1^				
Farm I	17			
Farm II	18			

^1^ Only Warmblood horses 4–16 years included.

**Table 2 animals-11-01762-t002:** Phyla, classes and families with a relative abundance of >1% in the faecal microbiota.

Phylum	Class	Family	Relative Abundance (%)
Bacteroidetes			50.1
	Bacteroidia		50.1
		Rikenellaceae	12.9
		p-251-o5	11.7
		Prevotellaceae	9.6
		F082	4.1
		Bacteroidales_UCG_001	3.4
		Bacteroidales_RF16	2.3
Firmicutes			28.4
	Clostridia		22.8
		Lachnospiraceae	9.8
		Oscillospiraceae	3.5
		UCG-010	2.5
		Ruminococcaceae	1.6
		Anaerovoracaceae	1.2
	Bacilli		3.0
		Erysipelatoclostridiaceae	1.2
	Negativicutes		2.6
		Acidaminococcaceae	2.2
Spirochaetes			7.1
	Spirochaetia		6.7
		Spirochaetaceae	6.7
Verrucomicrobia			6.5
	Kiritimatiellae *		6.0
Fibrobacteres			5.0
	Fibrobacteria		5.0
		Fibrobacteraceae	5.0
Cyanobacteria			1.0
	Vampirivibrionia		1.0

* Further classification down to family level currently unavailable.

## Data Availability

The datasets supporting this article have been uploaded to the sequence read archive as part of the Appendix A and are available under accession PRJEB44895.

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
