# Peer review of "The Equine Faecal Microbiota of Healthy Horses and Ponies in The Netherlands: Impact of Host and Environmental Factors"

_animals, 2021, doi:10.3390/ani11061762_

Round 1

Reviewer 1 Report

The study by Theelen et al. studied the impact of environmental factors on fecal microbiota of horses and ponies

It is an interesting and valuable study to evaluate many environmental factor on equine microbiota at a time.

Minor revision should be done:

  • Line 362, Faecalibacterium prauznitzii should be printed in italics.
  • Line 382 describes Acidaminococcaceae, Rikenellaceae and P-251-o5 were less abundant in ponies compared to horses. But line 383 describes Acidaminococcaceae, Saccharimonadaceae, Rikenellaceae, Spirochaetaceae and P-251-o5 were more abundant in ponies.

Author Response

Reviewer comment: The study by Theelen et al. studied the impact of environmental factors on fecal microbiota of horses and ponies. It is an interesting and valuable study to evaluate many environmental factor on equine microbiota at a time.

Author response: We would like to thank the reviewer for the kind words and the time and effort to review our manuscript. The comments have been addressed to the best of our ability. See below for a point-by-point response.

Reviewer comment: Line 362, Faecalibacterium prauznitzii should be printed in italics.

Author response: Adjusted as requested (line 371)

Reviewer comment: Line 382 describes Acidaminococcaceae, Rikenellaceae and P-251-o5 were less abundant in ponies compared to horses. But line 383 describes Acidaminococcaceae, Saccharimonadaceae, Rikenellaceae, Spirochaetaceae and P-251-o5 were more abundant in ponies.

Author response: For Acidaminococcaceae, Rikenellaceae and P-251-o5, we observed that specific genera/species within these families increase or decrease according to horse type, suggesting that bacterial shifts occur within these families. A statement to explain this observation has been added to the manuscript. (lines 389-396)

Reviewer 2 Report

To have such a large number of subjects for the research may have led to unwieldy data but I think it adds a lot of power to your results. Your observations in section 4.2 are very important in explaining the disparity in the various equine microbiome research. I would like to see more detail in some places in the article. The abstract needs work. More information in the Introduction would be helpful. I have put my comments into a Word document after converting the pdf for you to go over (sorry the formatting gets a bit messed up with the conversion)..

Author Response

Reviewer comment: To have such a large number of subjects for the research may have led to unwieldy data but I think it adds a lot of power to your results. Your observations in section 4.2 are very important in explaining the disparity in the various equine microbiome research. I would like to see more detail in some places in the article. The abstract needs work. More information in the Introduction would be helpful. I have put my comments into a Word document after converting the pdf for you to go over (sorry the formatting gets a bit messed up with the conversion).

Author response: We would like to thank the reviewer for the kind words and the time and effort to review our manuscript. The comments from the word document have been addressed to the best of our ability. See below for a point-by-point response.

Reviewer comment M1: Abstract needs work -more on methodology of your sampling. Also, there is nothing in the abstract about the part of research where you validated the fecal sample collection timing.

Author response: More details on methodology of sampling have been added to the abstract as requested. Also, a statement on the validation of this sampling technique (results of the pilot study investigating the effect of air exposure on fecal microbiota composition) has been added to the abstract. (lines 30-34)

Reviewer comment M2: I think it is hard to extrapolate much beyond the farms themselves because the samples came from only 2 farms, even though there were a lot of horses and ponies on the one farm, but it does give a base-line for real-world situations.

Author response: Agreed, a statement reflecting this concern has been added to the limitations section of the manuscript (lines 482-487)

Reviewer comment M3: What were the housing conditions for these 18 horses?

Author response: This information has been added to the manuscript as requested (lines 81-82)

Reviewer comment M4: Were all 61 horses/ponies fed the same concentrate? What type of concentrate was it? High starch? High fiber? High fat? To do a valid comparison of < or > 2 kg concentrate it would need to be the same concentrate or at least the same type (ie. low starch, etc). A little more information on the concentrate would be helpful

Author response: Unfortunately, not all horses were fed the same type of concentrates. We do realize this comparison is therefore rather crude. The text in the manuscript regarding concentrates has been adjusted to provide some more detail and to clearly address this limitation (lines 83 and 425-426)

Reviewer comment M5: Were all the horses sampled both summer and winter? If only half were sampled in each season and if there was a significant effect, then doesn’t that confound the results of all the other comparisons?

Author response: Given the study design and the fact that we used a heterogeneous study population, all factors included in this study potentially affect one another (including season). One of the goals of this study was to identify these factors of influence on the faecal microbiota (so these can be taken into account when designing future studies). By including a larger number of horses and ponies, we have tried to increase power and limit the potential effect of one factor on the other. We also made sure all (other) factors were distributed evenly between groups for each comparison (by factor). The factors ‘season of sampling’ and ‘pasture access’ seemed to be correlated (as horses more frequently have pasture access during summer months) and this was, therefore, addressed in the discussion. (lines 456-459)

Reviewer comment M6: Do you think the freezing temperatures that the feces were exposed to before collection could have had anything to do with the differences in beta-diversity?

Author response: Environmental temperatures were > 0°C at the time of sample collection. Environmental temperatures at the moment of sample collection have been added to the manuscript. (lines 85-86)

Reviewer comment M7: fibrolytic

Author response: Adjusted as requested. (Line 390)

Reviewer comment M8: Make this wording clearer because it looks on first reading that these three families are both less and more abundant in ponies.

Author response: The text has been edited to prevent confusion for the readers as suggested. (lines 389-396)

Reviewer comment M9: Did you look at the starch and fiber content of the concentrates? It seems like >2 kg & <2kg is too arbitrary given the variation in fiber content of today’s concentrates unless all the horses/ponies were fed the same type of concentrate. Please add more details on types of concentrates the horses were receiving.

Author response: Unfortunately, not all horses were fed the same type of concentrates. We do realize this comparison is therefore rather crude. The text in the manuscript regarding concentrates has been adjusted to provide some more detail and to address this limitation (lines 83 and 425-426)

Reviewer comment M10: Article title should not be capitalized.

Author response: Edited as requested (lines 531-532)

Reviewer comment M11: Article title should not be capitalized.

Author response: Edited as requested (lines 538-539)

Reviewer comment M12: Capitalize Thoroughbred.

Author response: Edited as requested (line 631)